# THE LOGIC OF RATIONAL GRAPH NEURAL NETWORKS

## ABSTRACT

The expressivity of Graph Neural Networks (GNNs) can be described via appropriate fragments of the first-order logic. In this context, uniform expressivity guarantees that a GNN can express a logical query without the parameters depending on the size of the input graphs. It has been established that the two-variable guarded fragment with counting (GC2) can be expressed uniformly via Rectified Linear Unit (ReLU) GNNs Barceló et al. (2020). Moreover, GC2 is the fragment that can be expressed at most by a GNN with any activation function. In this article, we prove that, on the contrary of ReLU GNNs, there are GC2 queries that cannot be uniformly expressed via any GNN with rational activations. As a consequence, non-polynomial activation functions do not grant GNNs GC2 uniform expressivity in general, answering an open question formulated by Grohe (2021). We then present a strict subfragment of GC2 (RGC2), and prove that rational GNNs can express RGC2 queries uniformly over all graphs. Our numerical experiments illustrates that despite this theoretical disadvantage, rational GNNs are still able to learn some GC2 queries if some level of error is allowed.

## 1 INTRODUCTION

Graph Neural Networks (GNNs) are deep learning architectures for input data that incorporates some relational structure represented as a graph, and have proven to be very performant and efficient for various types of learning problems ranging from chemistry Reiser et al. (2022), social network analysis Zhang et al. (2022), bioinformatics and protein-ligand interation Khalife et al. (2021); Knutson et al. (2022), autonomous driving Singh & Srivastava (2022); Gammelli et al. (2021); Chen et al. (2021), and techniques to enhance optimization algorithms Khalil et al. (2017; 2022) to name a few. Understanding the ability of GNNs to compute, approximate or express functions over graphs, and the dependence of their capacity on the activation function is beneficial for every aspect of learning: from the understanding of the target class, the design of a GNN for a given task, to the algorithmic training. For example, certain activation functions may endow GNNs with more expressivity than others, or require less parameters to express the same functions.

In this context, several approaches have been conducted in order to describe and characterize the expressive power of GNNs. The first approach consists in comparing GNNs to other standard computation models on graphs such as the *color refinement* or the *Wesfeiler-Leman* algorithms. These comparisons stand to reason, because the computational models of GNNs, Weisfeiler-Leman, and color refinement algorithms are closely related. They all operate under the paradigm of inferring global graph structure from local neighborhood computations. In that regard, it has been proven Morris et al. (2019); Xu et al. (2018) that the color refinement algorithm precisely captures the expressivity of GNNs. More precisely, there is a GNN distinguishing two nodes of a graph if and only if color refinement assigns different colors to these nodes. This results holds if one supposes that the size of the underlying neural networks are allowed to grow with the size of the input graph. Hence, in his survey, Grohe (2021) emphasizes the fact that this equivalence has been established only for unbounded GNN, and asks: *Can GNNs with bounded size simulate color refinement?* In Aamand et al. (2022), the authors answer by the negative if the underlying neural network are supposed to have Rectified Linear Unit (ReLU) activation functions. In Khalife & Basu (2023) the authors provide a generalization of this result, for GNNs with piecewise polynomial activation functions. Furthermore, explicit lower bounds on the neural network size to simulate the color refinement can be derived for piecewise-polynomial activation functions given upper bounds on the number of regions of a neural network with piecewise-polynomial activation.

The second line of research to study the expressive power of GNNs is to characterize the types of boolean queries that a GNN can simulate. For example, can a GNN express if a vertex of a graph is part of a clique of given size? Furthermore, can we characterize the set of queries that can be expressed by GNNs? If the number of parameters of the GNN does not depend on the size of the input graph, the GNN is said to express the query *uniformly*. Therefore, uniform expressivity guarantees that the number of parameters remains only dependent on the complexity of the target query[1]. This becomes relevant from a practical standpoint as it captures the expressivity of GNNs with a fixed number of parameters with respect to the number of vertices in the input graphs. The suitable logic over labelled graphs for GNNs is a two variable fragment of *graded model logic*, referred to as GC2. Any GNN expresses a query of this logic, and conversely, any query of this logic can be expressed by a GNN whose size and iterations only depends on the depth of the query Barceló et al. (2020); Grohe (2021). For specific activation functions such as ReLUs, the size of a GNN required to express a given query of GC2 does not depend on the size of the input graph, but only on depth of the query. In recent results Grohe (2023), the author provides a non-uniform description of the logic expressible by GNNs with rational piecewise-linear activations (or equivalently, rational ReLUs). The non-uniform results are presented for GNNs with general arbitrary real weights and activation functions. Additionally, Rosenbluth et al. (2023) compared the impact of the aggregation function on the expressivity of GNNs, showing that GNNs with Max or Mean aggregation functions have distinct expressivity from the Sum aggregation GNNs. In this article, we focus on uniform expressivity and consider the following question: *What is the impact of the activation on the logic uniformly expressed by GNNs?*

A natural start for such investigation are polynomial activations. They have demonstrated clear limitations for feedforward Neural Networks (NNs) as exposed with the celebrated theorem of approximation Leshno et al. (1993), stating that polynomial activations are the *only ones* leading to NNs being unable to approximate any continuous function on a compact set. For example, in the case of NNs, rational activations (i.e. fractions of polynomials) do not share this limitation. Beyond this ability, rational activations yield efficient approximation power of continous functions Boullé et al. (2020), if one is allowed to consider different rational activation functions (of bounded degree) in the neural networks. In the same spirit, this article exposes a comparison of the power of the activation function in the case of GNNs. In particular, we will compare rational activations with those of piecewise linear activations whose expressive power is known.

**Main contributions.** In this work we present an additional step towards a complete understanding of the impact of the activation function on the logical expressivity of GNNs. We show that the class of GNNs with *rational activations* (i.e. all activation functions that are fractions of polynomials[2]) have weaker expressivity than piecewise linear activations, or ReLUs. We prove that GNNs with rational activations cannot express all GC2 queries uniformly over all graphs, while they can with ReLU GNNs. Our approach demonstrates that this limitation is inherent to rational activations, as our findings remain valid even when the following contidions are allowed: i) the weights of the rational GNNs are arbitrary real numbers with infinite precision, and ii) the weights of the ReLU GNNs are restricted to integers (also, the underlying neural networks are supposed to have finitely many linear pieces). This result holds for sum-aggregation function and can be extended to aggregation functions that are rational with bounded-degree[3]. This shows how the power of GNNs can change immensely if one changes the activation function of the neural networks. These results also seem to suggest that ReLU GNNs possess special ability for uniform expressivity over rational GNNs, a property in contrast with the efficient approximation power of rational NNs Boullé et al. (2020). In addition, we describe a strict subfragment of GC2, called RGC2, that rational GNNs can express uniformly.

We would like to point out we focus our study solely on the ability of classes of GNNs to express given queries (i.e. can we even find a GNN in the class that does the job?). In particular, we do not address the question of how the learning process impacts expressivity in this work, but we briefly touch upon those interactions with some numerical experiments.

---

[1] We shall see that we can attribute a notion of depth over queries, that can be interpreted as a measure of complexity.

[2] For theoretical and practical considerations, we limit our study to fractions having *no real pole*.

[3] This is abuse of the notion "rational" as aggregation functions are defined on multisets. Here, an aggregation is rational if it remains rational in the entries of the multiset.

The rest of this article is organized as follows. Section 2 presents the definitions of GNNs and the background logic. In Section 3, we state our main result and compare it to the existing ones. Section 4 presents an overview of the proof of our negative result. Section 5 presents the technical definitions and overview for our positive results. Our numerical experiments are presented in Section 6. We conclude with some discussion and open questions in Section 7.

## 2 PRELIMINARIES

### 2.1 RATIONAL GRAPH NEURAL NETWORKS (GNNS)

We assume the input graphs of GNNs to be finite, undirected, simple, and vertex-labeled: a graph is a tuple $G = (V(G), E(G), P_1(G), \cdots, P_\ell(G))$ consisting of a finite vertex set $V(G)$, a binary edge relation $E(G) \subset V(G)^2$ that is symmetric and irreflexive, and unary relations $P_1(G), \cdots, P_\ell(G) \subset V(G)$ representing $\ell > 0$ vertex labels. In the following, we suppose that the $P_i(G)$'s form a partition of the set of vertices of $G$, i.e. each vertex has a unique label. Also, the number $\ell$ of labels, which we will also call *colors*, is supposed to be fixed and does not grow with the size of the input graphs. This allows to model the presence of features of the vertices of input graphs. In order to describe the logic of GNNs, we also take into account access to the color of the vertices into the definition of the logic considered, as we shall see in Section 2.2. For a graph $G$ and a vertex $v \in V(G)$, $N_G(v) := \{y \in V(G) : \{x, y\} \in E\}$ is the set of neigbhors of $v$. If there is no ambiguity about which graph $G$ is being considered, we simply use $N(v)$. $|G|$ will denote the number of vertices of $G$. We use simple curly brackets for a set $X = \{x \in X\}$ and double curly brackets for a multiset $Y = \{\{y \in Y\}\}$. For a set $X$, $|X|$ is the cardinal of $X$. When $m$ is a positive integer, $\mathfrak{S}_m$ is the set of permutations of $\{1, \cdots, m\}$. $\|.\|$ is the Euclidean norm, i.e., for a vector $x \in \mathbb{R}^m$, $\|x\| := \left(\sum_{i=1}^m x_i^2\right)^{1/2}$. Finally, if $E$ is a real vector space, $I$ a subset of $E$, $\mathsf{span}\{I\}$ refers to the set of all finite linear combinations of vectors of $I$, i.e. $\mathsf{span}\{I\} := \{\sum_{i=1}^m \lambda_i x_i : \lambda \in \mathbb{R}^m, x_1, \cdots, x_m \in I, m \in \mathbb{N} - \{0\}\}$.

**Definition 1** (Rational fractions/functions). *For a positive integer $m$, $\mathbb{R}(X_1, \cdots, X_m)$ refers to the field of rational fractions over the field $\mathbb{K} = \mathbb{R}$. For any positive integer $m$, a rational fraction is a pair $(P, Q)$ (represented as $\frac{P}{Q}$), where $P, Q \in \mathbb{R}[X_1, \cdots, X_m]$ are multivariate polynomials. The degree of $R$ is the pair $(\deg(P), \deg(Q))$. In the following, we make no formal distinction between rational fractions and rational functions, defined as functions taking values as the ratio between two polynomial functions. We also restrict our GNN study to fractions having no real pole, i.e. the polynomial $Q$ has no root in $\mathbb{R}^m$.*

**Definition 2** (Neural Network (NN)). *Fix an activation function $\sigma : \mathbb{R} \to \mathbb{R}$. For any number of hidden layers $k \in \mathbb{N}$, input and output dimensions $w_0, w_{k+1} \in \mathbb{N}$, a $\mathbb{R}^{w_0} \to \mathbb{R}^{w_{k+1}}$ Neural Network (NN) with activation function $\sigma$ is given by specifying a sequence of $k$ natural numbers $w_1, w_2, \cdots, w_k$ representing widths of the hidden layers and a set of $k + 1$ affine transformations $T_i : \mathbb{R}^{w_{i-1}} \to \mathbb{R}^{w_i}$, $i = 1, \ldots, k + 1$. Such a NN is called a $(k + 1)$-layer NN, and has $k$ hidden layers. The function $f : \mathbb{R}^{w_0} \to \mathbb{R}^{w_{k+1}}$ computed or represented by this NN is:*

$$f = T_{k+1} \circ \sigma \circ T_k \circ \cdots T_2 \circ \sigma \circ T_1.$$

In the following, the *Rectified Linear Unit* activation function $\mathrm{ReLU} : \mathbb{R} \to \mathbb{R}_{\geq 0}$ is defined as $\mathrm{ReLU}(x) = \max(0, x)$. The *Sigmoid* activation function $\mathrm{Sigmoid} : \mathbb{R} \to (0, 1)$ is defined as $\mathrm{Sigmoid}(x) = \frac{1}{1+e^{-x}}$.

**Definition 3** (Graph Neural Network (GNN)). *A GNN is characterized by:*

- *A positive integer $T$ called the number of iterations, positive integers $(d_t)_{t \in \{1, \cdots, T\}}$ and $(d'_t)_{t \in \{0, \cdots, T\}}$ for inner dimensions. $d_0 = d'_0 = \ell$ is the input dimension of the GNN (number of colors) and $d_T$ is the output dimension.*

- *a sequence of combination and aggregation functions $(\mathsf{comb}_t, \mathsf{agg}_t)_{t \in \{1, \cdots, T\}}$. Each aggregation function $\mathsf{agg}_t$ maps each finite multiset of vectors of $\mathbb{R}^{d_{t-1}}$ to a vector in $\mathbb{R}^{d'_t}$. For any $t \in \{1, \cdots, T\}$, each combination function $\mathsf{comb}_t : \mathbb{R}^{d_{t-1}+d'_t} \longrightarrow \mathbb{R}^{d_t}$ is a neural network with given activation function $\sigma : \mathbb{R} \longrightarrow \mathbb{R}$.*

*The update rule of the GNN at iteration $t \in \{0, \cdots, T-1\}$ for any labeled graph $G$ and vertex $v \in V(G)$, is given by:*

$$\xi^{t+1}(v) = \mathsf{comb}_t(\xi^t(v), \mathsf{agg}_t\{\{\xi^t(w) \, : \, w \in \mathcal{N}_G(v)\}\})$$

*Each vertex $v$ is initially attributed an indicator vector $\xi^0(v)$ of size $\ell$, encoding the color of the node $v$: the colors being indexed by the palette $\{1, \cdots, \ell\}$, $\xi^0(v) = e_i$ (the $i$-th canonical vector) if the color of the vertex $v$ is $i$. We say that a GNN has rational activations provided the underlying neural network* comb *has rational activation functions.*

**Remark 1.** *The type of GNN exposed in Definition 3 is sometimes referred to as* aggregation-combine *GNNs* without global readout. *Here are a few variants that can be found in the litterature:*

- Recurrent *GNNs, where* $\mathsf{comb}_t$ *and* $\mathsf{agg}_t$ *functions do not depend on the iteration $t$. The results presented in this article extends to recurrent GNNs as any aggregation-combine GNN without global readout can be reduced polynomially to a recurrent one.*

- *GNNs* with global readout, *for which aggregation functions also take as input the embeddings of all the vertices of the graph. See Remark 3 for known results from a logic standpoint.*

- *General* Message-passing *GNNs that allow operations before the aggregation on the neighbors as well as the current vertex (*targeted *messages). We refer to Grohe & Rosenbluth (2024) for a some elements of comparison of expressivity of targeted vs standard ones.*

### 2.2 Logical background

**First order logic on graphs.** In this subsection we present the logical foundations for queries in graph theory. We refer the interested reader to Appendix B containing details for the general construction. Let $\ell > 0$ be a fixed number of colors, and let $G = (V(G), E(G), P^1(G), ..., P^\ell(G))$ be a colored graph. The first-order language of graph theory we consider is built up in the usual way from a *alphabet* containing:

- the logical connectives $\wedge, \vee, \neg, \rightarrow$
- the quantifiers $\forall$ and $\exists$
- equality symbol $=$
- the *universe $A$* of the logic is given by $A := V$
- variables $x_0, x_1, \cdots$ (countably many)
- the *vocabulary $S$* is composed of:
    - a binary edge relation symbol $E$: $(x, y) \in A^2$ are related if and only if $(x, y) \in E$.
    - unary relation symbols $\mathsf{Col}_1, \cdots, \mathsf{Col}_\ell$ indicating if a vertex has a given color

The set of *formulas* in the logic is a set of strings over the alphabet. To interpret these formulas and the logic over every graph, we need for each graph a map $I$ defined on the relations:

- for every $i \in [\ell]$, $I(\mathsf{Col}_i) : A \rightarrow \{0, 1\}$
- $I(E) : A \times A \rightarrow \{0, 1\}$

The pair $(A, I)$ is called an *$S$-structure* for the first order logic FO(S). Provided this $S$-structure, one can safely construct simple examples of formulas at the *graph level* (we will see in the next paragraph that we need something more to interpret them at the *vertex level*). Namely, the following formula interpreted over a graph $G = (V, E)$ expresses that no vertex $v \in V$ is isolated: $\forall x \exists y E(x, y)$. Similarly, the formula $\forall x \neg E(x, x)$ expresses the fact that we do not want any self loops. A more interesting example is given by

$$\psi := \forall x \forall y [E(x, y) \rightarrow E(y, x) \wedge x \neq y] \tag{1}$$

expressing that $G$ is undirected and loop-free. Similarly

$$\phi := \forall x \exists y \exists z (\neg(y = z) \wedge E(x, y) \wedge E(x, z)) \wedge \forall w (E(x, w) \rightarrow ((w = y) \vee (w = z))) \tag{2}$$

expresses that every node $x$ of the considered graph has exactly two out-neigbhors.

**Free variables and assignments.** Since GNNs can output values for every vertex of a graph, the formulas we are interested in order to describe them from a logical standpoint must also take as "input" some vertex variable. Therefore, we need to add some component to the $S$-structure described as above, resulting in what we call an *interpretation*.

First we need the concept of *bound* and *free* variable of a formula. A bound variable of a formula is a variable for which the scope of a quantifier applies. In comparison, a free variable is not bound to a quantifier in that formula. Previous Formulas 1 and 2 contain no free variable. In contrast, the formula $\phi(x) := \exists y \exists z (\neg(y = z) \land E(x,y) \land E(x,z))$, which expresses that vertex $v$ has two out-neighbors, has a single free variable $x$. To interpret formulas with one (resp. $k$) free variable, we need an *assignment* that maps the set of free variables in the logic, to the universe $A$ (resp. $A^k$).

Now, an *interpretation* of FO(S) is a pair $(\mathcal{U}, \beta)$ where $\mathcal{U}$ is an $S$-structure and $\beta$ is an assignment. To interpret a formula with free variables, every graph is associated to an $S$-structure and an assignment; both are usually implicit in practice. Any formula in FO(S) (with and without free variables) can now be thought of as a $0/1$ function on the class of all interpretations of FO(S):

1. If $\phi$ is a formula without a free variable (in this case, the formula is said to be a *sentence*), then any graph $G$ is an $S$-structure and is mapped to $0$ or $1$, depending on whether $G$ satisfies $\phi$ or not.

2. If $\phi(x)$ is a formula with a single free variable $x$, then any pair $(G, v)$, where $G = (V, E)$ is a graph and $v \in V$, is an interpretation with $G$ as the $S$-structure and the assignment $\beta$ maps $x$ to $v$. Thus, every pair $(G, v)$ is mapped to $0$ or $1$, depending on whether $(G, \beta)$ satisfies $\phi$ or not. This example can be extended to handle formulas with multiple free variables, where we may want to model $0/1$ functions on subsets of vertices.

**Definition 4.** *(Queries as Boolean functions) Let $G$ be a graph and let $v \in V(G)$ be a vertex of $G$. If $Q$ has a free variable query (of type (2) above), $Q(G, v) \in \{0, 1\}$ refers to the query interpreted and evaluated using the pair $(G, v)$.*

**Definition 5.** *The* depth *of a formula $\phi$ is defined recursively as follows. If $\phi$ is of the form $\mathsf{Col}_i$ for $i \in [\ell]$, then its depth is $1$. If $\phi = \neg\phi'$ or $\phi = \forall x \phi'$ or $\phi = \exists x \phi'$, then the depth of $\phi$ is the depth of $\phi'$ plus 1. If $\phi = \phi_1 \star \phi_2$ with $\star \in \{\lor, \land, \to, \leftrightarrow\}$, then the depth of $\phi$ is $1 + \max(\mathsf{depth}(\phi_1), \mathsf{depth}(\phi_2))$.*

In order to characterize the logic of GNNs, we are interested in a fragment of the first order logic, defined as follows.

**Definition 6** (Graded (or guarded) model logic with counting (GC) and GC2 Barceló et al. (2020))**.** *The alphabet of GC2 is composed of*

- *the logical connectives $\land, \lor, \neg, \to$*

- *the quantifiers $\forall$, and for every positive integer $N$, $\exists^{\geq N}$*

- *the universe $A = V$*

- *variables $x_0, x_1, \cdots$ (countably many)*

- *the vocabulary $S$:*

  - *a binary edge relation symbol $E$: $(x, y) \in A^2$ are related if and only if $(x, y) \in E$.*
  - *unary relation symbols $\mathsf{Col}_1, \cdots, \mathsf{Col}_\ell$ indicating if a vertex has a given color*

*In contrast with the first order logic, we do not have access to equality (=). $\exists$ is simply $\exists^{\geq 1}$. For a given unary relation $R$, the quantifier $\exists^{\geq N} x R(x)$ means that there exists at least $N$ elements $x$ verifying relation $R$. Similarly, $\exists^{\geq N} x E(x, y)$ means that at least $N$ vertices adjacent to $y$ in the considered graph. GC-formulas are formed from the atomic formulas by the Boolean connectives and quantification restricted to formulas of the form $\exists^{\geq p} y(E(x, y) \land \psi))$, where $x$ and $y$ are distinct variables and $x$ appears in $\psi$ as a free variable. Note that every formula of GC has at least one free variable. For example, $\phi(x) := \neg(\exists^{\geq 2} y(E(x, y) \land \mathsf{Col}_1(y)) \land \exists^{\geq 3} z(E(x, z) \land \mathsf{Col}_2(z)))$ is a GC formula.*

We refer to the 2-variable fragment of GC as GC2 (i.e. formulas with only two variables). Equivalently, a GC2 formula $F$ is either $\mathsf{Col}_i(x)$ (returning 1 or 0 for one of the palette colors) or one of the following:

$$\neg\phi(x), \quad \phi(x) \wedge \psi(x), or \quad \exists^{\geq N} y(E(x,y) \wedge \phi(y))$$

where $N$ is a positive integer and $\phi$ and $\psi$ are GC2 formulas of lower depth than $F$.

**Example 1** (Barceló et al. (2020)). *All graded modal logic formulas naturally define unary queries. Suppose $\ell = 2$ (number of colors), and $\mathsf{Col}_1 = Red$, $\mathsf{Col}_2 = Blue$. Let:*

$$\gamma(x) := Blue(x) \wedge \exists y(E(x,y) \wedge \exists^{\geq 2} x(Edge(y,x) \wedge Red(x))$$

*$\gamma$ queries if $x$ has blue color, and has at least one neigbhor which has at least two red neighbors. Then $\gamma$ is in GC2. Now,*

$$\delta(x) := Blue(x) \wedge \exists y(\neg E(x,y) \wedge \exists^{\geq 2} x E(y,x) \wedge Red(x))$$

*is not in GC2 because the use of the guard $\neg E(x,y)$ is not allowed. However,*

$$\eta(x) := \neg(\exists y(E(x,y) \wedge \exists^{\geq 2} x E(y,x) \wedge Blue(x))$$

*is in GC2 because the negation $\neg$ is applied to a formula in GC2.*

**Definition 7.** *Suppose that $\xi$ is the vertex embedding computed by a GNN. We say that a GNN expresses uniformly a unary query $Q$ there is a real $\epsilon < \frac{1}{2}$ such that for all graphs $G$ and vertices $v \in V(G)$.*

$$\begin{cases} \xi(G,v) \geq 1 - \epsilon & \text{if } v \in Q(G) \\ \xi(G,v) \leq \epsilon & \text{if } v \notin Q(G) \end{cases}$$

## 3 FORMAL STATEMENTS OF RESULTS

Given Definition 7, we are now equipped to state the known previous results regarding the expressivity of GNNs:

**Theorem 1.** *Barceló et al. (2020); Grohe (2021) Let $Q$ be a unary query expressible in graded modal logic GC2. Then there is a GNN whose size depends only on the depth of the query, that expresses $Q$ uniformly.*

**Remark 2.** *Let $\ell$ be the number of colors of the vertices in the input graphs, the family of GNNs with $\mathsf{agg} = \mathsf{sum}$, and $\mathsf{comb}(x,y) = \mathrm{ReLU}(Ax + By + C)$ (where $A \in \mathbb{N}^{\ell \times \ell}$, $B \in \mathbb{N}^{\ell \times \ell}$ and $C \in \mathbb{N}^\ell$) is sufficient to express all queries of GC2 uniformly. This result follows from the constructive proof of Theorem 1 in Appendix A. Furthermore, for each query $Q$ of depth $q$, there is a GNN of this type with at most $q$ iterations that expresses $Q$ uniformly.*

**Example 2.** *Let $Q$ be the following GC2 query:*

$$Q(x) := Red(x) \wedge (\exists y E(x,y) \wedge Blue(y))$$

*asking if the vertex $x$ has red color, and if it has a neighbor with blue color. Writing the subformulas of $Q$: $sub(Q) = (Q1,Q2,Q3,Q4)$ with $Q_1 = Red$, $Q_2 = Blue$, $Q_3 = \exists(E(x,y) \wedge Q_2(y)$, and $Q_4 = Q = Q_1 \wedge Q_3$, let*

$$A = \begin{pmatrix} 1 & 0 & 0 & 0 \\ 0 & 1 & 0 & 0 \\ 0 & 0 & 0 & 0 \\ 1 & 0 & 1 & 0 \end{pmatrix}, B = \begin{pmatrix} 0 & 0 & 0 & 0 \\ 0 & 0 & 0 & 0 \\ 0 & 1 & 0 & 0 \\ 0 & 0 & 0 & 0 \end{pmatrix}, c = \begin{pmatrix} 0 \\ 0 \\ 0 \\ -1 \end{pmatrix}$$

*and let $\sigma$ be the clipped ReLU function, i.e. $\sigma(\cdot) := \min(1, \max(0, \cdot))$ (the clipped ReLU can be computed by a neural network with ReLU activations). Then, it can be verified that $Q$ can be computed in 4 iterations with the update rule:*

$$\xi^0(G,v) = 1, \quad \xi^{t+1}(G,v) := \sigma(A\xi^t(G,v) + B(\sum_{w \in N_G(v)} \xi^t(w)))$$

*i.e. $Q_i(G,v) = \xi^4(G,v)_i$. In particular, $Q_i(G,v) = 1 \iff \xi^4(G,v)_i = 1$. The ability of GNNs to compute exactly GC2 queries is used in the proof of Theorem 1. We emphasize here that one cannot mimic the proof for sigmoid activations, even by replacing exact computation by uniform expressivity.*

In general, the logic of GNNs and their different variants beyond first-order queries remains elusive. However, for aggregation-combine GNNs without global readout (cf. Remark 1), Theorem 1 has the following partial converse:

**Theorem 2.** *Barceló et al. (2020) Let Q be a unary query expressible by a GNN and also expressible in first-order logic. Then Q is expressible in GC2. Furthermore, a logical classifier is captured by an aggregation-combine GNN without global readout if and only if it can be expressed in GC2 logic.*

**Remark 3.** *The logic that fits GNNs* with global readout *is not GC2. If $C2$ is the fragment of the first order logic with counting quantifiers ($\exists^{\geq p}$) and with at most 2 variables; then we have the following result Barceló et al. (2020):* Let Q be a Boolean or unary query expressible in C2. Then there is a GNN with global readout that expresses Q.

In contrast with Theorem 1, we prove:

**Theorem 3.** *There are GC2 queries that no GNN with rational activations can uniformly express.*

Equivalently, if $\mathcal{L}_R$ (resp. $\mathcal{L}_{ReLU}$) is the set of first order logical queries uniformly expressible by GNNs with rational activations (resp. ReLU activations), then $\mathcal{L}_R \subsetneq (\mathcal{L}_{ReLU} = \text{GC2})$. The query used in our proof uses logical negation:

$$Q_p(s) := \neg \left( \exists^{\geq 1} x (E(s,x) \wedge \exists^{\leq (p-1)} s E(x,s)) \right)$$

and can be extended to a large family of queries (cf. Remark 5). We can obtain the following corollary by immediate contradiction, as GNNs with rational activations and aggregations can simulate logical negation:

**Corollary 1.** *There are queries of GC2 using only the guarded existential quantifiers with counting $\exists^{\geq K} E$, the logical and $\wedge$ and the atomic formulas Col(.), that GNNs with rational activations cannot uniformly express.*

We complete the negative result describe above via a description of a strict subfragment of GC2, RGC2 (presented in Section 5 ) that rational GNNs can uniformly express:

**Theorem 4.** *For any query Q of RGC2, there exists a rational GNN that expresses Q uniformly over all graphs.*

## 4 RATIONAL GNNS HAVE LIMITED EXPRESSIVITY

**Overview.** To prove our result we construct a GC2 query $Q$ that no GNN with rational activation can express over all graphs. We prove this statement by contradiction: on the one hand we interpret the embedding returned by a rational GNN on a set of given input graphs, as a rational function of some parameters of the graph structure. On the other hand, we interpret $Q$ on the same set of input graphs. We show that if GNN were to uniformly express $Q$, then the rational function obtained by the first evaluation cannot verify the constraints imposed by $Q$. Our approach and can easily extend to a large family of GC2 queries.

Similarly to those considered in Khalife & Basu (2023), our set of inputs are formed using rooted unicolored trees of the form shown in Figure 1 which is a tree of depth two whose depth one vertices have prescribed degrees $k_1, \cdots, k_m$, with $k_1, \cdots, k_m \geq 0$. We first collect three elementary Lemmas, one that will be useful to *extract* monomials of largest degree in a multivariate polynomial (Lemma 1) then used for rational fractions (Lemma 2). Since the trees are parameterized by $m$-tuples of integers $k_1, \ldots, k_m$, the embedding of the root node computed by the GNN at any iteration is a function of these $m$ integers. Since the activations are rational, these embeddings of the root node are multivariate symmetric rational functions of $k_1, \ldots, k_m$ (Lemma 3). Furthermore, the degree of these rational functions is bounded by a constant independent of $m$. Our proof of Theorem 3 builds on these results combined with fundamental properties of symmetric polynomials and rational functions.

**Remark 4.** *Note that the proof of Theorem 1 can be extended to a larger family of queries. Namely, for any integer $p \geq 2$, let*

$$Q_p(s) := \neg \left( \exists^{\geq 1} x (E(s,x) \wedge \exists^{\leq (p-1)} s E(x,s)) \right) = \forall x E(s,x) \exists^{\geq p} s E(x,s)$$

$Q_p$ *queries if vertex $s$ has neighbors whose degree are all at least $p$. Then any $(Q_p)_{p \in \mathbb{N}}$ cannot be expressed by any GNN with rational activations.*

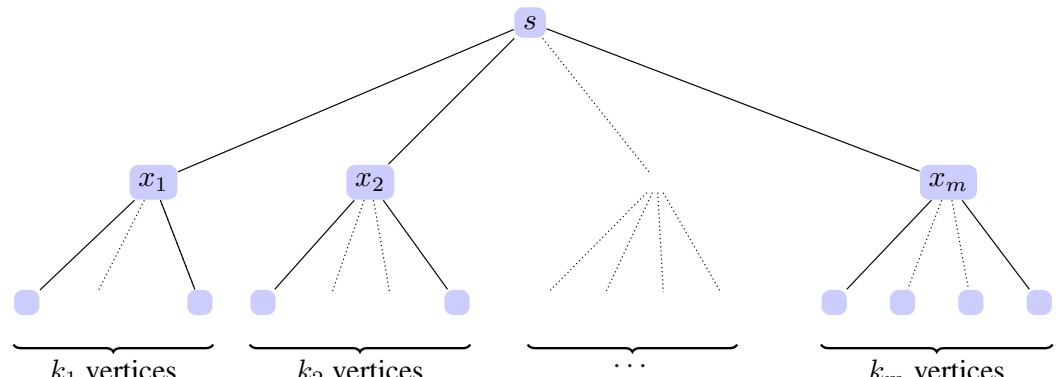

Figure 1: $T[k_1, \cdots, k_m]$

**Remark 5.** *The proof of Theorem 3 was initially attempted using queries of the form:*

$$\tilde{Q}_p(s) = \neg \left( \exists^{\geq 1} x (E(s,x) \wedge \exists^{\geq (p+1)} s E(x,s)) \right) = \forall x E(s,x) \exists^{\leq p} s E(x,s)$$

*Which expresses that all the neighbors of $s$ have degree at most $p$. Note the similarity between $Q_p$ from Remark 5 and $\tilde{Q}_p$. Although $\tilde{Q}_p$ also seems a good candidate that cannot be expressed uniformly by a GNN with rational activations and aggregations, we could not conclude with the same approach as in the proof of Theorem 3, due to the following interesting fact:*

*There exists $\epsilon > 0$ and a sequence of symmetric polynomial $(p_m)_{m \in \mathbb{N}} \in \mathbb{R}[x_1, \cdots, x_m]$ of bounded degree (i.e. there exists an integer $q$ such that for any $m$, $\deg(p_m) \leq q$) and for any $m$, $p_m$ is greater than $\epsilon$ on the vertices of the unit hypercube $\{0,1\}^m$, and less than $-\epsilon$ on all the other points of $\mathbb{N}^m$. $p_m = 1 - \sum_{i=1}^{m} x_i^2 + \sum_{i=1}^{m} x_i^4$ is an example of such sequence of symmetric polynomials.*

## 5 TOWARDS A RATIONAL FRAGMENT OF GC2

We now turn our attention towards a subfragment of GC2 that use of existential quantifiers aligned in the "same direction", at the exception of the very last quantifier, as negation will only be allowed for the last subformula. In particular, this removes logical conjunctions and negations inside nested subformulas. This fragment will be used to describe what rational GNNs can express at the very least. Informally, such limitation arises from the aggregation phase when the messages in the neighborhood of a node, one obtains a signal that can become unbounded, and we lose track of the number of neighbors that verify a given query, except at the very first iteration (captured by the set $\Omega_0$ in the Definition below). Our counterexamples (see Remark 5) confirm this is indeed happening.

**Definition 8** (Fragment RGC2 $\subseteq$ GC2). *The fragment RGC2 is composed of logical queries of $\Omega$ constructed as follows:*

- *$\Omega_0$ contains $\mathsf{Col}_i$, $\neg\mathsf{Col}_i$, and $\exists^{\geq K} y E(x,y) Col_i(y)$ for some $i \in [\ell]$ and for some $K \in \mathbb{N}$.*

- *$\Omega+ := \{\mathcal{H}^{(m)}(\phi) : m \in \mathbb{N}, \phi \in \Omega_0\}$, where $\mathcal{H} : \phi \mapsto \tilde{\phi}$ extends queries to 1-hop neighborhoods, i.e. $\tilde{\phi}(x) := \exists^{\geq 1} y (E(x,y) \wedge \phi(y))$.*

- *$\Omega := \{\neg\psi \text{ with } \psi \in \Omega_+\} \cup \Omega_+$.*

A few comments are in order. Note that the counting quantifier $\exists^{\geq K}$ with $K > 1$ is not allowed on top of other guarded fragments, so

$$\phi_1(x) := \exists^{\geq 3} y E(x,y)(\exists^{\geq 1} x E(x,y) \wedge Red(x)) \quad \phi_2(x) := \exists^{\geq 1} y E(x,y)(\exists^{\geq 1} x E(x,y) \wedge Red(x))$$

$\phi_1$ is not in RGC2, but $\phi_2$ is.

Similarly,
$$\phi_1(x) = \neg(\exists^{\geq 1} y E(x,y)(\exists^{\geq 1} x E(y,x)(\exists^{\geq 1} y E(x,y) \wedge Red(x))))$$

belongs to RGC2,

$$\phi_2(x) = \neg(\exists^{\geq 1} y E(x,y)(\exists^{\geq 1} x E(y,x)(\exists^{\leq 1} y E(x,y) \wedge Red(x))))$$

does not (alternates between existential and non existential quantifiers). Neither does

$$\phi_3(x) = \neg(\exists^{\geq 1} y E(x,y)(\neg\exists^{\geq 1} x E(y,x)(\exists^{\geq 1} y E(x,y) \wedge Red(x))))$$

as it alternates between $\exists^{\geq 1}$ and $\neg\exists^{\geq 1}$ in the nested formulas.

Finally, unlike the query $Q_1$ in Remark 5

$$\phi_4(x) := \forall y E(x,y)(\exists^{\leq 1} E(x,y)) = \neg(\exists^{\geq 1} y E(x,y)(\exists^{\geq 1} x E(x,y)))$$

is in RGC2.

**Proof overview.** Our positive result that GNNs with rational activations can express GC2 uniformly (formally stated in Theorem 4) mostly relies on two observations. The first observation (stated in Lemma 4 of the appendix), is that given any activation function of degree $\geq 2$, and for any polynomial, there exists a NN with that activation function that computes $P$. The second observation is that in order to express RGC2 queries that are in $\Omega_0$, we only require to be able to set a combine function to 0 for a *finite number* of values (integers), and at least one on the other integers. This is achievable via a NN using the first observation and interpolation, provided the activation function is polynomial of degree at least 2. Then, we then construct a proof by induction on the depth of the queries of RGC2, starting with queries in $\Omega_0$, and then generalizing to all queries of RGC2.

## 6   NUMERICAL EXPERIMENTS

In this section, we investigate if the limitations of rational GNNs on the uniform side impacts the ability of rational GNNs to learn GC2 queries with some level of error. To do so, we consider the following queries:

$$Q_1(v) := \neg\left(\exists^{\geq 1} y (E(y,v) \wedge (\neg\exists^{\geq 2} v E(v,y)))\right) = \forall y \left(E(y,v) \wedge \exists^{\geq 2} z E(z,y)\right)$$

$Q_1(v)$ is expressing that all neigbhors of $v$ have degree at least two. Note that $Q$ is in GC2 and has depth 4 (here, since trees are unicolored, we removed the color atomic queries for the sake of presentation. Otherwise, the depth of the query would be 5).

$$Q_2(v) = \mathsf{Red}(v) \wedge \left(\exists^{\geq 1} x E(x,v) \wedge \left(\exists^{\geq 1} v E(v,x) \wedge \mathsf{Blue}(v)\right) \wedge \left(\exists^{\geq 1} v E(v,x) \wedge \mathsf{Red}(v)\right)\right)$$

$Q_2(v)$ is expressing that $v$ is red, and has a neighbor that has a red neighbor and a blue neighbor. $Q_2$ is in GC2 as well and has depth 7. The vertices of the trees are colored as follows: the source and depth-one vertices are red, and only the leaves are blue.

We compare the GNN's ability to learn GC2 queries, depending on the activation considered (Rational vs. clipped reLU (CReLU): $x \mapsto \min(1, \max(0, x))$). We consider the same two queries as above and train two distinct GNNs: (a) A first GNN with rational activations, and (b) A second GNN with CReLU activation functions. Both GNNs have 4 and 7 iterations when trained to learn the first and second query respectively. Each iteration is attributed his own combine function, a feedforward neural network with one hidden layer. This choice is justified by Theorem 1 that guarantees one can compute exactly a GC2 query (with the ReLU one) with the number of iterations corresponding to the depth of the query, and with only one hidden layer for the combine function. Our training dataset is composed of 3750 graphs of varying size (between 50 and 2000 nodes) of average degree 5, generated randomly. Our testing dataset is composed of 1250 graphs with varying size between 50 and 2000 vertices, of average degree 5.

The experiments were conducted on a Mac-OS laptop with an Apple M1 Pro chip. The neural networks were implemented using PyTorch 2.3.0 and Pytorch geometric 2.2.0, as well as the Rational activations libary Delfosse et al. (2020). The details of the implementation are provided in the supplementary material.

**Results.** Both GNNs seem to generalize well for both instance as the mean square error stabilizes around a value for large graphs. Furthermore, the Rational GNN is even better at learning the first query than the CReLU GNN. We explain this phenomenon as follows. Theorem 1 only guarantees

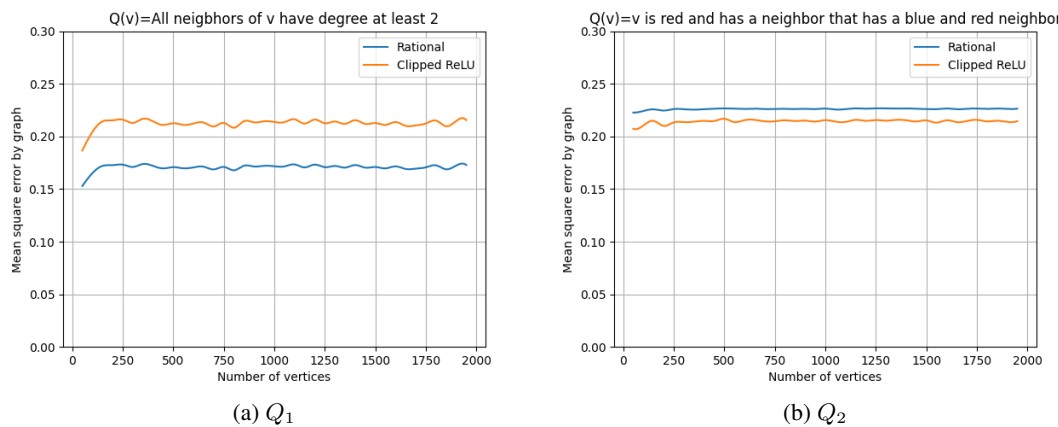

(a) $Q_1$                      (b) $Q_2$

Figure 2: Learning queries with Rational GNN vs CReLU GNN. The mean square error per graph on the test set is displayed as a function of the order of the graph.

that there exists a set of weights such that a CReLU GNN expresses uniformly the query. However, this result do not say anything about how difficult it is to reach those weights, starting from random weights and using a variant of stochastic gradient descent. For example, the weights of the matrices of the GNNs that achieve this can be chosen with $\{-1, 0, 1\}$ entries and with at most two non zero entries per row. It seems unlikely that these set of weights can be reached easily with our method of training the GNNs. The numerical procedure to train those GNNs largely impact the reachability of those weights, limiting the theoretical advantage of CReLU over rational GNNs.

## 7 DISCUSSION AND OPEN PROBLEMS

Analyzing rigorously how the expressivity of GNNs is altered with the choice of activation function is valuable as it can help in selecting an appropriate architecture for a given learning problem. In this regard, the universal approximation properties of neural networks, and in particular the efficiency of rational neural networks Boullé et al. (2020) for that purpose, may lead to believe that rational GNNs have maximal expressivity from a formal logical standpoint. Our results show that it is not the case, and rational GNNs have strictly weaker expressivity than ReLU GNNs. However, it is unclear if such limitation carries over for other (non rational) aggregation functions such as max. Furthermore, the proof of our positive result does not show that negation is *never* allowed in subformulas. We conjecture it is not the case and that RGC2 is close to the logic expressed by rational GNNs. An interesting path for future research is to understand whether this desirable property of ReLU GNNs confer them a significant advantage over rational GNNs on graphs of bounded order, which may be more relevant for practical applications.

It is also essential to note that expressivity is just one facet of practical use of GNNs and the related machine learning algorithms. This article does not delve into other crucial aspects such as the GNN's ability to generalize from provided data, and the computational efficiency of learning and inference. In particular, we have not investigated the ability of a GNN to learn a logical query from examples and how the numerical optmization part used for training may impact expressivity. Our numerical experiments suggest that these factors may be at play, including the architecture chosen from learning that may differ from the ones that allow to express the uniform queries. We also wish to convey that theoretical investigations on the expressivity of GNNs and logical expressivity can suggest potential avenues to integrate logic-based and statistical reasoning in GNN architectures.

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

## A   PROOF THAT PIECEWISE LINEAR GNNS (AND AGG=SUM) ARE AS EXPRESSIVE AS GC2

*Proof.* We will prove the following claim from which the Theorem 1 will be an immediate corollary.

*Claim. Let $Q$ be a query in GC2, and let $sub(Q) = (Q_1, Q_2, \cdots, Q_d)$ be an enumeration of the subformulas of $Q$. Then, there exists a ReLU GNN returning $\xi^t$ such that for graph $G$ and any vertex $v \in V(G)$, $\xi^t(G, v) \in \{0, 1\}^d$, and for any $i \in \{1, \cdots, d\}$, $\xi_i^{t+1}(G, v) = 1 \iff Q_i(G, v) = 1$.*

The overall GNN will take as input the graph $G$ as well as for each node of $v$, $\xi^0(G, v) \in \{0, 1\}^\ell$ encoding the colors of each node of $G$; and after $d$ iterations, outputs for each node a vector $\xi^d(G, v) \in \{0, 1\}^d$. Furthemore, at each intermediate iteration $t \in \{1, \cdots, d-1\}$, the constructed GNN will verify $\xi^t(G, v) \in \{0, 1\}^d$. This property will be crucial for the inductive argument to go through.

In order to prove the claim, we simply need to find appropriate $(\mathsf{comb}_t)_{1 \le t \le d}$ and $(\mathsf{agg}_t)_{1 \le t \le d}$ functions such that if $\xi^t$ verifying the update rule:

$$\xi^{t+1}(G, v) = \mathsf{comb}_t(\xi^t(G, v), \mathsf{agg}_t(\{\{\xi^t(G, v)) : v \in \mathcal{N}_G(v)\}\}))$$

then $\xi^t$ computes the given query $Q$. We will prove that we can find such $\mathsf{comb}_t$ and $\mathsf{agg}_t$ functions by induction on the depth of $Q$.

**Base case**. If $Q$ has depth 1, $Q$ is one of $\ell$ color queries, and this can be computed via a GNN in one iteration, whose underlying neural network is the projection onto the $i-$th coordinate, i.e.

- $\mathsf{comb}_0(x, y) = \mathrm{proj}_i(x)$

- $\mathsf{agg}_0$ can be chosen as any aggregation function.

**Induction step.** Let suppose $Q$ be a query of depth $d > 1$. By the induction hypothesis, we here suppose that we have access to some $(\mathsf{comb}_t)_{1 \le t \le d-1}$ and $(\mathsf{comb}_t)_{1 \le t \le d-1}$ such that for any $j \in \{1, \cdots, d-1\}$ and for any $1 \le i \le j$, $Q_j(v) = 1 \iff \xi_j^i(v) = 1$. Recall that $sub(Q) = (Q_1, Q_2, \cdots, Q_d)$, i.e. the $Q_i$s form an enumeration of the subformulas of $Q$. In particular, $Q_d = Q$. In the following construction, $\xi^i$ keeps "in memory" the output of the subformulas $Q_j$, for $j \le i$. For each case described below, we show that we are able to construct $\mathsf{comb}_d$ and $\mathsf{agg}_d$ in the following form:

- $\mathsf{comb}_d : \mathbb{R}^d \times \mathbb{R}^d \to \mathbb{R}^d, (x, y) \mapsto \sigma(A_d X + B_d Y + c_d)$, where $\sigma(\cdot) = \min(1, \max(0, \cdot))$ is the clipped ReLU. Note that a clipped ReLU can be computed by a Neural Network with ReLU activations.

- $\mathsf{agg}_d$ is the sum function.

Due to the inductive nature of GC2, one of the following holds:

- Case 1: there exist subformulas $Q_j$ and $Q_k$ of $Q$, such that $\ell(Q_j) + \ell(Q_k) = d$ and $Q = Q_j \wedge Q_k$

- Case 2: $Q(x) = \neg Q_j(x)$ where $Q_j$ is a query of depth $d - 1$

- Case 3: there exists a subformula $Q_j$ of $Q$ such that $\ell(Q_j) = d - 1$ and $Q(x) = \exists^{\ge N} y(E(x, y) \wedge Q_j(y))$

We will first give general conditions on the update of the combinations and aggregate functions, and then conclude that these conditions are actually be met by constant comb and agg functions:

- $A_d$ gets the same first $d-1$ rows as $A_{d-1}$.

- $B_d$ gets the same first $d-1$ rows as $B_{d-1}$

- $c_d$ get the same first $d-1$ coordinates as $c_{d-1}$,

- **Case 1:** The $d$-th row of $A_d$ gets all zeros except: $(A_d)_{jd} = 1$, $(A_d)_{kd} = 1$. The $d$-th row of $B_t$ is set to 0. Set $(c_d)_d = -1$.

- **Case 2:** The $d$-th row of $A_d$ gets all zeros except: $(A_d)_{jd} = -1$. The $d$-th row of $B_d$ is set to 0. Set $(c_d)_d = 1$.

- **Case 3:** The $d$-th row of $A_d$ is set to 0. The $d$-th row of $B_d$ is set to 0 except $(B_d)_{jd} = 1$. Set $(c_d)_d = -N + 1$.

What remains to prove is the following:

$$\text{For any } i \in \{1, \cdots, d\}, \quad \xi_i^d(G, v) = 1 \iff Q_i(G, v) = 1$$

Due to the update rule, the first $d-1$ coordinates of $\xi^d(G, v)$ are the same as $\xi^{d-1}(G, v)$. Hence, the property is true for $i \leq d-1$ by immediate induction. We are left to show that $\xi_d^d(G, v) = 1 \iff Q(G, v) = Q_d(G, v) = 1$. Here, we use the fact that:

- for every node $v$, every coordinate of $\xi^{d-1}(v)$ is in $\{0, 1\}$.

- $\sigma$ is the clipped ReLU activation function: $\sigma(\cdot) = \min(1, \max(0, \cdot))$.

Since $\xi^d(G, v) = \sigma(A_{d-1}\xi^{d-1}(G, v) + B_{d-1}\sum_{w \in N(v)} \xi^{d-1}(G, w) + c_{d-1})$ This follows from an immediate discussion on the three cases described previously, and ends the induction on $d$.

An important feature of the update described above is that $(A_d, B_d, c_d)$ verifies all the conditions imposed for every $(A_t, B_t, c_t)_{1 \leq t \leq d}$ to compute all subformulas $Q_t$. The updates of $A_t$, $B_t$ and $c_t$ are only made for the $t$-th row and $t$-th entry, and do not depend on the previous columns but only on the query $Q$. Hence, we may as well start from the beginning by setting $(A_t, B_t, c_t)$ to $(A_d, B_d, c_d)$, instead of changing these matrices at every iteration $t$. In these conditions, the combination function $\text{comb}_t$, parametrized by $A_t$, $B_t$ and $c_t$ can be defined independently of $t$. The same holds for $\text{agg}_t$ as it can be chosen as the sum for any iteration. 

$\square$

**Remark 6.** *The proof of Grohe (2021) presents an approach where the ReLU GNN is non-recurrent (each $\text{comb}_t$ in that case depends on $t$). The fact that $\xi^t$ is a $\{0, 1\}$-vector is also crucial so that the argument goes through. In particular, both proofs do not extend to other activations (such as sigmoid), as there is no function $f : \mathbb{R} \to \mathbb{R}$ such that for some $0 < \epsilon < \frac{1}{2}$, and for any $x_1, \cdots, x_N \in [0, \frac{1}{2} - \epsilon] \cup [\frac{1}{2} + \epsilon, 1]$,*

$$f(\sum_{i=1}^N x_i) \geq \frac{1}{2} + \epsilon \iff \text{there are at least } p \ x_i\text{'s such that } x_i \geq \frac{1}{2} + \epsilon$$

*On the one hand, it is necessary that $f(x) \geq \frac{1}{2} + \epsilon$ for any $x \geq p$. Furthermore, it is possible to pick $x_1, \cdots, x_N$ such that for any $i$, $x_i \in [0, \frac{1}{2} - \epsilon]$ but $x_1 + \cdots x_N \geq p$. This in turn would imply $f(\sum_{i=1}^N x_i) \geq \frac{1}{2} + \epsilon$ but all $x_i$'s are smaller than $\frac{1}{2}$. The previous property becomes verified if one restricts to $\{0\} \cup \{1\}$ and $f_p(.) = cReLU(\cdot - p + 1)$ where cReLU is the clipped ReLU.*

# B  LOGIC BACKGROUND: GENERAL DEFINITIONS

**Definition 9.** *A first order logic is given by a countable set of symbols, called the* alphabet *of the logic:*

1. *Boolean connectives:* $\neg, \vee, \wedge, \rightarrow, \leftrightarrow$

2. *Quantifiers:* $\forall, \exists$

3. *Equivalence/equality symbol:* $\equiv$

4. *Variables:* $x_0, x_1, \ldots$ *(finite or countably infinite set)*

5. *Punctuation:* $(, )$ *and* $,$.

6. (a) *A (possibly empty) set of* constant *symbols.*
   (b) *For every natural number $n \geq 1$, a (possibly empty) set of $n$-ary function symbols.*
   (c) *For every natural number $n \geq 1$, a (possibly empty) set of $n$-ary relation symbols.*

**Remark 7.** *Items 1-5 are common to* any *first order logic. Item 6 changes from one system of logic to another. Example: In Graph theory, the first order logic has:*

- *no constant symbols*

- *no function symbol*

- *a single 2-ary relation symbol $E$ (which is interpreted as the edge relation between vertices). When graphs are supposed labeled with $\ell$ colors: $\ell$ function symbols $\mathsf{col}_1, \cdots, \mathsf{col}_\ell$. $\mathsf{col}_i(v \in G)$.*

*The set of symbols from Item 6 is called the* vocabulary *of the logic. It will be denoted by $S$ and the first order logic based on $S$ will be denoted by $FO(S)$.*

**Definition 10.** *The set of* terms *in a given first order logic $FO(S)$ is a set of strings over the alphabet defined inductively as follows:*

1. *Every variable and constant symbol is a term.*

2. *If $f$ is an $n$-ary function symbol, and $t_1, \ldots, t_n$ are terms, then $f(t_1, \ldots, t_n)$ is a term.*

**Definition 11.** *The set of* formulas *in a given first order logic is a set of strings over the alphabet defined inductively as follows:*

1. *If $t_1, t_2$ are terms, then $t_1 \equiv t_2$ is a formula.*

2. *If $R$ is an $n$-ary relation symbol, and $t_1, \ldots, t_n$ are terms, then $R(t_1, \ldots, t_n)$ is a formula.*

3. *If $\phi$ is a formula, then $\neg\phi$ is a formula.*

4. *If $\phi_1, \phi_2$ are formulas, then $(\phi_1 \vee \phi_2)$, $\phi_1 \wedge \phi_2$, $\phi_1 \rightarrow \phi_2$ and $\phi_1 \leftrightarrow \phi_2$ are formulas.*

5. *If $\phi$ is a formula and $x$ is a variable, then $\forall x\phi$ and $\exists x\phi$ are formulas.*

The set of all variable symbols that appear in a term $t$ will be denoted by $\mathrm{var}(t)$. The set of *free variables in a formula* is defined using the inductive nature of formulas:

1. $\mathrm{free}(t_1 \equiv t_2) = \mathrm{var}(t_1) \cup \mathrm{var}(t_2)$
2. $\mathrm{free}(R(t_1, \ldots, t_n)) = \mathrm{var}(t_1) \cup \ldots \cup \mathrm{var}(t_n)$
3. $\mathrm{free}(\neg\phi) = \mathrm{free}(\phi)$
4. $\mathrm{free}(\phi_1 \star \phi_2) = \mathrm{var}(\phi_1) \cup \mathrm{var}(\phi_2)$, where $\star \in \{\vee, \wedge, \rightarrow, \leftrightarrow\}$
5. $\mathrm{free}(\forall x\phi) = \mathrm{free}(\phi) \setminus \{x\}$
6. $\mathrm{free}(\exists x\phi) = \mathrm{free}(\phi) \setminus \{x\}$

**Remark 8.** *The same variable symbol may be a free symbol in $\phi$, but appear bound to a quantifier in a subformula of $\phi$.*

**Definition 12.** *The set of* sentences *in a first order logic are all the formulas with no free variables, i.e., $\{\phi : \mathrm{free}(\phi) = \emptyset\}$.*

**Definition 13.** *Given a first order logic $FO(S)$, an $S$-structure is a pair $\mathcal{U} = (A, I)$ where $A$ is a nonempty set, called the* domain/universe *of the structure, and $I$ is a map defined on $S$ such that*

    *1. $I(c)$ is an element of $A$ for every constant symbol $c$.*

    *2. $I(f)$ is a function from $A^n$ to $A$ for every $n$-ary function symbol $f$.*

    *3. $I(R)$ is a function from $A^n$ to $\{0, 1\}$ (or equivalently, a subset of $A^n$) for every $n$-ary relation symbol $R$.*

*Given an $S$-structure $\mathcal{U} = (A, I)$ for $FO(S)$, an* assignment *is a map from the set of variables in the logic to the domain $A$. An* interpretation *of $FO(S)$ is a pair $(\mathcal{U}, \beta)$, where $\mathcal{U}$ is an $S$-structure and $\beta$ is an assignment.*

We say that an interpretation $(\mathcal{U}, \beta)$ *satisfies* a formula $\phi$, if this assignment restricted to the free variables in $\phi$ evaluates to 1, using the standard Boolean interpretations of the symbols of the first order logic in Items 1-5 of Definition 9.

**Definition 14.** *The* depth *of a formula $\phi$ is defined recursively as follows. If $\phi$ is of the form in points 1. or 2. in Definition 11 , then its depth is 1. If $\phi = \neg\phi'$ or $\phi = \forall x\phi'$ or $\phi = \exists x\phi'$, then the depth of $\phi$ is the depth of $\phi'$ plus 1. If $\phi = \phi_1 \star \phi_2$ with $\star \in \{\vee, \wedge, \rightarrow, \leftrightarrow\}$, then the depth of $\phi$ is one plus the maximum of the depths of $\phi_1$ and $\phi_2$.*

*This is equivalent to the depth of the tree representing the formula, based on the inductive definition. The* length/size *of the formula is the total number nodes in this tree. Up to constants, this is the number of leaves in the tree, which are called the* atoms *of the formula.*

## C   ADDITIONAL DEFINITION AND LEMMAS

**Definition 15** (Embeddings and refinement). *Given a set $X$, an embedding $\xi$ is a function taking as input a graph $G$ and a vertex $v \in V(G)$, and returns an element $\xi(G, v) \in X$. We say that an embedding $\xi$ refines an embedding $\xi'$ if and only if for any graph $G$ and any $v \in V(G)$, $\xi(G, v) = \xi(G, v') \implies \xi'(G, v) = \xi'(G, v')$. When the graph $G$ is clear from context, we use $\xi(v)$ as shorthand for $\xi(G, v)$.*

**Definition 16** (Color refinement). *Given a graph $G$, and $v \in V(G)$, let $(G, v) \mapsto \mathsf{col}(\mathsf{G}, \mathsf{v})$ be the function which returns the color of the node $v$. The color refinement refers to a procedure that returns a sequence of embeddings $cr^t$, computed recursively as follows:*

*- $cr^0(G, v) = \mathsf{col}(G, v)$*

*- For $t \geq 0$, $\mathsf{cr}^{t+1}(G, v) := (\mathsf{cr}^t(G, v), \{\{\mathsf{cr}^t(G, w) : w \in N(v)\}\})$*

*In each round, the algorithm computes a coloring that is finer than the one computed in the previous round, that is, $\mathsf{cr}^{t+1}$ refines $\mathsf{cr}^t$. For some $t \leq n := |G|$, this procedure stabilises: the coloring does not become strictly finer anymore.*

The following connection between color refinement and GNNs will be useful to prove our main result. Notably, the theorem holds regardless of the choice of the aggregation function agg and the combination function comb.

**Theorem 5** (Morris et al. (2019); Xu et al. (2018)). *Let $d$ be a positive integer, and let $\xi$ be the output of a GNN after $d$ iterations. Then $\mathsf{cr}^d$ refines $\xi$, that is, for all graphs $G, G'$ and vertices $v \in V(G)$, $v' \in V(G')$, $\mathsf{cr}^{(d)}(G, v) = \mathsf{cr}^d(G', v') \implies \xi(G, v) = \xi(G', v')$.*

**Lemma 1.** *Let $p$ be a positive integer and let $S \subset \mathbb{N}^p$ be a finite subset of integral vectors of the nonnegative orthant, such that $S$ contains a non zero vector. Then there exist $x^* \in S$ and $u \in \mathbb{N}^p$ such that*

*i) if $|S| = 1$ then $\langle x^*, u \rangle > 0$*

*ii) if $|S| \geq 2$ then for any $x \in S - \{x^*\}$, $\langle x^*, u \rangle > \langle x, u \rangle$.*

*Proof.* If $|S| = 1$ the existence of $x^*$ and $u$ such that i) holds is clear as $S$ is the singleton of a vector that is non zero. To deal with ii) in the case $|S| \geq 2$, consider one vector $x^*$ maximizing the

Euclidean norm over $S$, i.e. $\|x^*\|^2 = \max_{x \in S}\|x\|^2$ then let $x \in S - \{x^*\}$. Such $x^*$ and $x$ exist because $S$ is finite and $|S| \geq 2$.

- Case 1: $x$ is not colinear to $x^*$. It follows from the Cauchy-Schwarz inequality, that $\langle x^*, x \rangle < \|x^*\|\|x\|$. Hence

$$\langle x^*, x^* - x \rangle = \|x^*\|^2 - \langle x^*, x \rangle > \|x^*\|(\|x^*\| - \|x\|) > 0$$

- Case 2: $x \in S - \{x^*\}$ is colinear to $x^*$, i.e. $x = \lambda x^*$ with $\lambda \in \mathbb{R}$. Since $x^*$ is maximizing the 2-norm on S, then $0 \leq \lambda < 1$. Then

$$\langle x^*, x^* - x \rangle = \|x^*\|(1 - \lambda) > 0$$

In both cases, $\langle x^*, x^* - x \rangle > 0$. Hence, we can set $u := x^* \in S \subset \mathbb{N}^p$, and the Lemma is proved. $\square$

The following Lemma simply states that the vector $u \in \mathbb{N}^p$ can be chosen the same if one is given a pair of sets $S$ and $S'$, at the price of one inequality being possibly non strict.

**Lemma 2.** *Let $p$ be a positive integer, and let $S, S'$ be two finite subsets of $\mathbb{N}^p$, such that $|S| \geq 2$ and $|S'| \geq 2$. Then there exists $u \in \mathbb{N}^p$, $x^* \in S$ and $y^* \in S$ such that for any $x \in S - \{x^*\}$ and any $y \in S' - \{y^*\}$ such that:*

*i) If there is $u \in \mathbb{N}^p$ maximizing the 2-norm both on $S$ and $S'$, i.e. $S$ and $S'$ have a common element $u = \arg\max(\{\|x\| : x \in S\}) = \arg\max(\{\|y\| : y \in S'\})$, then $\langle x^*, u \rangle > \langle x, u \rangle$ and $\langle y^*, u \rangle > \langle y, u \rangle$.*

*ii) If $S$ (resp. $S'$) has the element of strictly greatest 2-norm among $S \cup S'$, then $\langle x^*, u \rangle > \langle x, u \rangle$, $\langle y^*, u \rangle \geq \langle y, u \rangle$ and $\langle x^\star, u \rangle > \langle y^\star, u \rangle$ (resp. $\langle y^*, u \rangle > \langle x, u \rangle$, $\langle x^*, u \rangle \geq \langle y, u \rangle$ and $\langle y^\star, u \rangle > \langle x^\star, u \rangle$).*

*Proof.* Case i): In this case, both $u$ and $u'$ obtained from Lemma 1 coincide, as there is a common element maximizing the 2-norm over $S$ and $S'$. We know that there exists $u = \arg\max\{\|x\| : x \in S\} = u' = \arg\max\{\|y\| : y \in S'\} \in \mathbb{N}^p$ such that for any $x \in S - \{x^*\}$ and any $y \in S' - \{y^*\}$, $\langle x^*, u \rangle > \langle x, u \rangle$ and $\langle y^*, u \rangle > \langle y, u \rangle$.

Case ii): We only treat the case where $S$ has the strictly greatest 2-norm by symmetry of the role of $S$ and $S'$. Lemma 1 tells us there is $u := \arg\max(\{\|x\| : x \in S\})$ such that for any $x \in S - \{x^*\}$, $\langle x^*, u \rangle > \langle x, u \rangle$. Now, for every $y \in S'$, let $y = y_u + y_{u^\perp}$ be the orthonormal decomposition of $y$ in $\mathbb{R}^p = \mathsf{span}(u) \bigoplus u^\perp$. Note that both component vectors are still in $\mathbb{N}^p$.

For every $y \in S$, $\langle x^\star, y \rangle = \langle x^\star, y_u \rangle + \langle x^\star, y_{u^\perp} \rangle$. This proves that by selecting one $y^*$ with largest coordinate on $\mathsf{span}(u)$ gives: $\langle y^\star, u \rangle \geq \langle y, u \rangle$ for every $y \in S' - \{y^\star\}$, which is the inequality that was claimed. The inequality may not be strict as such $y^\star$ may not be the unique element maximizing $\langle y^\star, u \rangle$. The last claimed inequality follow from $x^* = u$ (cf. Lemma 1) and the chain of inequalities $\langle x^\star, u \rangle = \|x^\star\|^2 > \|y^\star\|\|x^\star\| \geq \langle y^\star, u \rangle$.

$\square$

**Lemma 3.** *Let $\xi^t(T[k_1, \ldots, k_m], s) \in \mathbb{R}^d$ be the embedding of the tree displayed in Figure 1 obtained via a GNN with rational activations after $t$ iterations, where $\xi^0(v) = 1$ for all vertices $v \in V(T[k_1, \ldots, k_m])$. Then, for any iteration $t$, and for every coordinate $\xi_i^t(T[k_1, \ldots, k_m], s)$, there exists a rational function $F_i$ such that $\xi_i^t(T[k_1, \ldots, k_m], s) = F_i(k_1, \cdots, k_m)$. Furthermore, the degrees of the numerator and denominator of each $F_i$ do not depend on $m$, but only on the underlying neural network and $t$.*

*Proof.* For clarity, we will perform two separate inductions, one for the existence of the rational function, and one for the degree boundedness.

**Rational function.** We first prove by induction on $t$ that, for any vertex $v \in V(T[k_1, \cdots, k_m])$, all the coordinates of $\xi^t(T[k_1, \ldots, k_m], v)$ are rational functions of the $k_i$'s.

Base case: for $t = 0$ this is trivial since all vertices are initialised with the constant rational function 1, whose degree does not depend on $m$.

Induction step: Suppose the property is true at iteration $t$, i.e for each node $w$, $\xi^t(T[k_1, \ldots, k_m], w)$ is (coordinate-wise) a rational functions of the $k_i$'s. Since

$$\xi^{t+1}(T[k_1, \ldots, k_m], v) = \mathsf{comb}_t(\xi^t(T[k_1, \ldots, k_m], v),$$
$$\mathsf{agg}_t(\{\{\xi^t(T[k_1, \cdots, k_m], w) : w \in N(v)\}\}))$$

where $\mathsf{comb}_t$ is a neural network with rational activations, hence a rational function. Also, $\mathsf{agg}_t$ is supposed rational in the entries of its multiset argument. Then by composition, each coordinate of $\xi^{t+1}(T[k_1, \ldots, k_m], v)$ is a rational function of $k_1, \cdots, k_m$.

**Degree boundedness.** We will prove that the degree of the numerators and denominators of $\xi^t(T[k_1, \cdots, k_m], s)$ are both respectively bounded by $q_t$ and $r_t$.

*Base case:* At the first iteration ($t = 0$), $P_m$ is constant equal to 1 ($q_1 = 1$ and $r_1 = 0$) for any $m$.

*Induction step:* Suppose that for any iteration $t \leq T$, there exists $q_t \in \mathbb{N}$ (that does not depend on $m$ nor the vertex $v \in V(T[k_1, \cdots, k_m])$), such that for every positive integer $m$, every vertex $v \in T[k_1, \cdots, k_m]$, and for every iteration $t$, $\deg(F_i) \leq (q_t, r_t)$. Then, using again the update rule:

$$\underbrace{\xi^{t+1}(T[k_1, \cdots, k_m], v)}_{Q_m} = \mathsf{comb}_t(\underbrace{\xi^t(T[k_1, \cdots, k_m], v)}_{R_m},$$
$$\underbrace{\mathsf{agg}_t(\{\{\xi^t(T[k_1, \cdots, k_m], w) : w \in N(v)\}\})}_{S_m})$$

$R_m$ and $S_m$ are rational fractions. By the induction hypothesis, for any $m$, $\deg(R_m) \leq (q_t, r_t)$ and $\deg(S_m) \leq (q_t, r_t)$.

Each coordinate of the function $\mathsf{comb}_t$ is a rational fraction of degree independent of $m$ (neural network with a rational activation). Let $(a_i, b_i)$ be the degree of its $i$-th coordinate for $i \in [d]$. The degree of the $i$-th coordinate $Q_m$ is at most $(a_i \times q_t r_t, b_i \times q_t r_t)$. Hence the property remains true at $t+1$ for each coordinate $i$, setting $q_{t+1} := (\max_{i \in [d]} a_i) \times q_t r_t$ and $r_{t+1} := (\max_{i \in [d]} b_i) \times q_t r_t$. $\square$

We can now build towards the:

*Proof of Theorem 3.* Recall that the fractions we consider have no real pole. Consider the following query of GC2:

$$Q(s) = \neg \left( \exists^{\geq 1} x (E(s, x) \wedge \exists^{\leq 1} s E(x, s)) \right) = \forall x E(s, x) \exists^{\geq 2} s E(x, s)$$

$Q$ is true if and only if all the neigbhors of the node $s$ have degree at least 2. Namely $Q(T[0, k_2, \cdots, k_m], s)$ is false and $Q(T[k_1, \cdots, k_m], s)$ is true for every positive integers $k_1, \cdots, k_m$. We will prove by contradiction that any bounded GNN with rational activations cannot uniformly express the query $Q$. Let $R_m := \xi^t(T[k_1, \cdots, k_m], s)$ be the embedding of the source node of $T[k_1, \cdots, k_m]$ returned by a GNN with rational activations, after a fixed number of iterations $t$.

Suppose that $R_m$ can uniformly express the query $Q$, then;

$$\begin{cases} R_m(k_1, \cdots, k_m) \geq 1 - \epsilon & \text{if } s \in Q(T[k_1, \cdots, k_m], s) \\ R_m(k_1, \cdots, k_m) \leq \epsilon & \text{if } s \notin Q(T[k_1, \cdots, k_m], s) \end{cases}$$

Let $\tilde{R}_m := R_m - \frac{1}{2}$ and $\epsilon' := \frac{1}{2} - \epsilon$. Interpreting the query $Q$ over $T[k_1, \cdots, k_m]$ implies the following constraints on the sequence of rational functions $\tilde{R}_m$:

$$\exists \epsilon' > 0 \text{ such that } \forall k \in \mathbb{N}^m \begin{cases} \exists i \in \{1, \cdots, m\}, k_i = 0 \implies \tilde{R}_m(k) \leq -\epsilon' \\ \forall i \in \{1, \cdots, m\}, k_i > 0 \implies \tilde{R}_m(k) \geq \epsilon' \end{cases} \quad (3)$$

Let $\tilde{R}_m = \frac{\tilde{P}_m}{\tilde{Q}_m}$ be the irreducible representation of $\tilde{R}_m$, and let $S$ (resp. $S'$) be the set of exponents of the monomials of $\tilde{P}_m$ (resp. $\tilde{Q}_m$), i.e.

$$S := \{(\alpha_1, \cdots, \alpha_m) \in \mathbb{N}^m : \alpha_1 + \cdots + \alpha_m \leq q \text{ and } k_1^{\alpha_1} \cdots k_m^{\alpha_m}$$
$$\text{is a monomial of } \tilde{P}_m\}$$

$$S' := \{(\alpha_1, \cdots, \alpha_m) \in \mathbb{N}^m : \alpha_1 + \cdots + \alpha_m \le q \text{ and } k_1^{\alpha_1} \cdots k_m^{\alpha_m}$$
$$\text{is a monomial of } \tilde{Q}_m\}$$

First $|S| = 0$ is impossible as $\tilde{R}_m$ would have no zeroes and Conditions 3 cannot be met. Henceforth we suppose that $|S| \ge 2$ and $|S'| \ge 2$ (the other cases are discussed hereafter).

Lemma 3 tells us that there exists a uniform bound on the degree of both $\tilde{P}_m$ and $\tilde{Q}_m$: there exists a positive integer $k$ such that for every integer $m$, $\deg(\tilde{P}_m) \le k$ and $\deg(\tilde{Q}_m) \le k$. Henceforth, we will suppose that $m > 2k$, so that $\max(\deg(\tilde{P}_m), \deg(\tilde{Q}_m)) < \frac{m}{2}$.

Consider the three exclusive cases:

i) there is a common element of $S$ and $S'$ maximizing the 2-norm

ii) $\max(\{\|x\| : x \in S\}) > \max(\{\|y\| : y \in S'\})$

iii) $\max(\{\|x\| : x \in S\}) < \max(\{\|y\| : y \in S'\})$

Lemma 2 (with $p = m$) tells us there exists $\alpha^* \in S$, $\beta^* \in S'$ and $u = (u_1, \cdots, u_m) \in \mathbb{N}^m$ such that

- Case i): for any $\alpha \in S - \{\alpha^*\}$ and for any $\beta \in S' - \{\alpha^*\}$, $\langle \alpha^*, u \rangle > \langle \alpha, u \rangle$ and $\langle \beta^*, u \rangle > \langle \beta, u \rangle$

- Case ii): for any $\alpha \in S - \{\alpha^*\}$ and for any $\beta \in S' - \{\alpha^*\}$, $\langle \alpha^*, u \rangle > \langle \alpha, u \rangle$ and $\langle \beta^*, u \rangle \ge \langle \beta, u \rangle$ and $\langle \alpha^*, u \rangle > \langle \beta^*, u \rangle$.

- Case iii): for any $\alpha \in S - \{\alpha^*\}$ and for any $\beta \in S' - \{\alpha^*\}$, $\langle \alpha^*, u \rangle > \langle \alpha, u \rangle$ and $\langle \beta^*, u \rangle \ge \langle \beta, u \rangle$ and $\langle \beta^*, u \rangle > \langle \alpha^*, u \rangle$.

***Claim 1:*** *In all cases, the (univariate) monomial $t^{\langle \alpha^*, u \rangle}$ is the monomial of $\tilde{P}_m(t^{u_1}, \cdots, t^{u_m})$ of largest degree.*

*Proof.*
$$\tilde{P}_m = \sum_{\alpha \in S} \gamma_\alpha k_1^{\alpha_1} \cdots k_m^{\alpha_m} \implies \tilde{P}_m(t^{u_1}, \cdots, t^{u_{m-1}}, t^{u_m}) = \sum_{\alpha \in S} \gamma_\alpha t^{\langle \alpha, u \rangle}$$

Hence, the monomial of largest degree of $\tilde{P}_m(t^{u_1}, \cdots, t^{u_m})$ is the one such that $\langle \alpha, u \rangle$ is (strictly) maximized when $\alpha \in S$. By construction, it is $\alpha^*$. $\qquad\square$

***Claim 2:*** *For every $\alpha \in S$, $\alpha' \in S'$, there exists $i \in [m]$ such that $\alpha_i = 0$ and $\alpha_i' = 0$.*

*Proof.* By contradiction: if there exists $\alpha \in S$ and $\alpha' \in S'$ such that for every $j \in [m]$, $\alpha_i > 0$ or $\alpha_i' > 0$, this would imply that $\max(\deg(\tilde{P}_m), \deg(\tilde{Q}_m)) \ge \frac{m}{2}$, a contradiction with the respective choice of $m$ and $k$ that gave $\max(\deg(\tilde{P}_m), \deg(\tilde{Q}_m)) < \frac{m}{2}$. $\qquad\square$

Without loss of generality, suppose that the index $i$ verifying Claim 2 for $\alpha^* \in S$ and $\beta^* \in S'$ is $i = m$.

***Claim 3:*** *In these conditions, the (univariate) monomial $t^{\langle \alpha^*, u \rangle}$ is also the monomial of of largest degree of $\tilde{P}_m(t^{u_1}, \cdots, t^{u_{m-1}}, 0)$.*

*Proof.* Evaluating $\tilde{P}_m$ in $(t^{u_1}, \cdots, t^{u_{m-1}}, 0)$ removes the contribution of each monomial of $P_m$ containing the last variable, and keeps only the contribution of the monomials containing it:
$$\tilde{P}_m = \sum_{\alpha \in S} \gamma_\alpha k_1^{\alpha_1} \cdots k_m^{\alpha_m} \implies \tilde{P}_m(t^{u_1}, \cdots, t^{u_{m-1}}, 0) = \sum_{\substack{\alpha \in S \\ \alpha = (\alpha_1, \cdots, \alpha_{m-1}, 0)}} \gamma_\alpha t^{\langle \alpha, u \rangle}$$

Therefore, the monomial of largest degree of $\tilde{P}_m(t^{u_1}, \cdots, t^{u_{m-1}}, 0)$ is the one such that $\langle \alpha, u \rangle$ is (strictly) maximized when $\alpha \in S$ and $\alpha_m = 0$. By construction, such $\alpha$ is $\alpha^*$. $\qquad\square$

Similarly, in case i) we obtain:

**Claim 4:** *The (univariate) monomial $t^{\langle \beta^*, u \rangle}$ is the monomial of $\tilde{Q}_m(t^{u_1}, \cdots, t^{u_m})$ of largest degree.*

**Claim 5:** *the (univariate) monomial $t^{\langle \beta^*, u \rangle}$ is also* the *monomial of* of largest degree of $\tilde{Q}_m(t^{u_1}, \cdots, t^{u_{m-1}}, 0)$.

Let $(\eta_\beta)_{\beta \in S'}$ be the coefficients of $\tilde{Q}_m$. In case i), the five claims combined imply that

$$\tilde{R}_m(t^{u_1}, \cdots, t^{u_{m-1}}, t^{u_m}) \underset{t \to +\infty}{\sim} \tilde{R}_m(t^{u_1}, \cdots, t^{u_{m-1}}, 0) \underset{t \to +\infty}{\sim} \frac{\gamma_{\alpha^*} t^{\langle \alpha^*, u \rangle}}{\eta_{\beta^*} t^{\langle \beta^*, u \rangle}}$$

In particular $\lim_{t \to +\infty} \tilde{R}_m(t^{u_1}, \cdots, t^{u_m}) = \lim_{t \to +\infty} \tilde{R}_m(t^{u_1}, \cdots, t^{u_{m-1}}, 0)$. This is a contradiction with Conditions 3.

If case ii) holds, then Claims 4) and 5) are not necessarily true as some exponents may cancel out. However, simply note in that case, due to strict inequalities obtained before Claim 1, that the degree of $\tilde{P}_m(t^{u_1}, \cdots, t^{u_m})$ is strictly greater than the one of $\tilde{Q}_m(t^{u_1}, \cdots, t^{u_m})$, implying that

$$\lim_{t \to +\infty} \tilde{R}_m(t^{u_1}, \cdots, t^{u_m}) = \lim_{t \to +\infty} \tilde{R}_m(t^{u_1}, \cdots, t^{u_{m-1}}, 0) \in \{-\infty, +\infty\}.$$

In case iii), with a similar reasoning as in case ii), we get $\lim_{t \to +\infty} \tilde{R}_m(t^{u_1}, \cdots, t^{u_m}) = \lim_{t \to +\infty} \tilde{R}_m(t^{u_1}, \cdots, t^{u_{m-1}}, 0) = 0$.

Finally, if $|S'| = 0$ then $\tilde{R}_m$ is a polynomial. In this case we do not need Claims 3 and 4, and the reasoning still applies. If $|S| = 1$ or $|S'| = 1$ then we can apply Lemma 1 (case i)) to the polynomials having only one monomial, and the same argument still goes through. $\qquad \square$

**Lemma 4.** *Let $\sigma$ be a univariate polynomial such that $\deg(\sigma) > 1$. Let $M > 1$ be an integer, and let $P \in \mathbb{R}[X]$ be a univariate polynomial of degree $M$. Then there exists a feedforward neural network $f$ with activation function $\sigma$, whose size depends only on $M$, such that $f = P$.*

*Proof.* Let $\sigma$ be a univariate polynomial such that $m := \deg(\sigma) > 1$. The set $\mathcal{A}_\sigma := \mathsf{span}\{\sigma_{w,\theta} : \mathbb{R} \to \mathbb{R}, \sigma_{w,\theta}(x) := \sigma(wx + \theta) : w \in \mathbb{R}, \theta \in \mathbb{R}\}$ is the set of polynomials of degree at most $m$. This can be deduced, for example, from the fact that the polynomials $1, \sigma(X), \sigma(X+1), \cdots, \sigma(X+m)$ are linearly independent, since the polynomials $1, (X+1)^m, (X+2)^m, \cdots, (X+m)^m$ are linearly independent (this is for example, a consequence of the via the positivity of the Vandermonde determinant for $m$ distinct integers). Hence, the family $\mathcal{F} = \{1, \sigma(X), \cdots, \sigma(X+m)\}$ form a basis of the vector space of polynomials of degree at most $m$. Since $\mathcal{A}_\sigma$ is the set of functions computed by a 1-hidden layer neural network with activation $\sigma$, we can prove the claim by induction on $M \geq 2$:

*Base case:* (M=2). Proved above.

*Induction step:* Suppose that for some integer $M$, for any integer $0 < i \leq M$, for any polynomial $P_i$ of degree $i$, there exists a neural network $f_M$ of size that depends only on $i$ such that $\tilde{f}_i = P_i$. Suppose we are given a polynomial $P$ of degree $M + 1$. By the induction hypothesis, we first reconstruct the polynomial $Q(X) := X^M$ as the output of one neuron. Now, $\sigma \circ Q$ has degree $M \times m > M$. Hence, $A_{\sigma Q}$ is the set of polynomials of degree at most $2m$. In particular, we can compute $P$ as the output of a neural network by adding an additional hidden layer.

$\qquad \square$

*Proof of Theorem 4.* Our proof is very similar to the one presented in Appendix A, and actually shows that a *polynomial* GNN will do the job. Let $Q$ be a query of RGC2 of depth $d$, that we decompose in subformulas $(Q_1, \cdots, Q_d)$. By definition, $Q_1 \in \Omega_0$ and all subqueries $Q_i$ for $i \in \{1, \cdots, d-1\}$ are in the positive fragment $\Omega_+$. Only $Q = Q_d$ is potentially in $\Omega$. We will prove by induction on $i \in \{1, \cdots, d-1\}$ that there is a polynomial GNN that outputs after $d$ iterations $\xi \in \mathbb{R}^d$ such that for every $i \in \{1, \cdots, d-1\}$ the coordinate $\xi_i$ is greater than 1 if the query $Q_i$ is verified (for the considered vertex and graph) and returns exactly 0 if the query is not verified. The subqueries belonging to $\Omega_0$ constitute our "base case", and we make standard induction on the depth when the query is in $\Omega_+ - \Omega_0$.

*Base case:* We first prove that every query of $\Omega_0$ can be expressed. Any query of $\Omega_0$ is composed of the following queries:

1) $Col_i$ or $\neg Col_i$ for some $i \in [\ell]$. Since the initial feature embedding $\xi^0(v, G) \in \{0,1\}^\ell$ encodes the color of the node $v$, one can construct a combine function that returns: i) in first coordinate 1 if the $i$-th coordinate is 1, and 0 when it is 0, and ii) zeroes in all $d-1$ coordinates. This insures that at the first iteration, the GNN expresses $Q_1$.

2) If the query is of the form $\exists^{\geq K} y E(x, y) Col_i(y)$ for some $i \in [\ell]$ and for some $K \in \mathbb{N}$. $\xi^1(v)$ is a $0-1$ vector constructed as in case 1) above. One can construct a polynomial function $P : \mathbb{R} \to \mathbb{R}$ (taking as input only the sum of the signals from the neigbhors) that is equal to 0 on every point of $\{0, \cdots, K-1\}$, and greater than 1 on $\mathbb{N} - \{0, \cdots, K-1\}$. This can be achieved, for instance, by interpolation with the polynomial $P = X(X-1)(X-2)\cdots(X-(K-1))$, and using Lemma 4. Concretely,

$$\xi^2(G, v) = \tilde{P}(\sum_{w \in N_G(v)} \xi^1(G, w))$$

where $\tilde{P}$ is a vector with coordinates that are polynomials. The second coordinate of $\tilde{P}$ (corresponding to the subquery considered) is $P$. The other coordinates of $\tilde{P}$ and the others are the identity function. It is easy to see that $\xi^2(G, v)$ has the desired property.

*Induction step:* For $i \geq 2$, Suppose we are given a subformula of $\Omega_+$, then $Q_i(x) = \exists^{\geq 1} y(E(x, y) \wedge Q_{i-1}(y))$ with $Q_{i-1} \in \Omega_+$. By the induction hypothesis, we can construct a GNN, whose output $\xi \in \mathbb{R}^d$ verifies: for every $j \in \{1, \cdots, i-1\}$, $\xi_j$ is at least 1 if $Q_j$ is verified and equal to 0 otherwise. Suppose we are given such GNN. At the update phase, when summing over the neighbors, one gets a signal equal to 0 if there is no neighbor verifying $Q_{i-1}$, and a signal of value at least 1 if one of them does. Simply using the combine function that returns the same signal for all the coordinates the $i$-th coordinate gives $\xi_i$ expressing $Q_i$. More precisely, we have

$$\xi^i(G, v) = A_i \xi^{i-1}(G, v) + B_i \sum_{w \in N_G(G, v)} \xi^{i-1}(G, w)$$

where the combine function is the identity map $\mathbb{R}^d \to \mathbb{R}^d$, $A_i \in \mathbb{R}^{d \times d}$ has same first $i-1$ rows as $A_{i-1}$ and other rows are set to zero. $B_i \in \mathbb{R}^{d \times d}$ has the same $i-1$ rows as $B_{i-1}$ and $i$-th row verifies $B_{i,i-1} = 1$. All first $i-1$ coordinates of $\xi^i$ remains the same, and the new $i$-th coordinate expresses the desired query.

Finally, if $Q = Q_d = \neg Q_{d-1}$ with $Q_{d-1} \in \Omega$. By the result obtained above, we are in possession of a polynomial GNN that is at least 1 when $Q_{d-1}$ is verified, and exactly 0 when not. Simply consider the update:

$$\xi^d(v, G) = A_d \xi^{i-1}(G, v)$$

Where the first $d-1$ rows of $A_d$ are the same as $A_{d-1}$ and the last row verifies $A_{d,d-1} = -1$. This insures that final query can be expressed by the GNN.

$\square$

