# OpenReview forum: "The logic of rational graph neural networks"
_ICLR.cc/2025/Conference — ICLR 2025 Conference Withdrawn Submission_

### Official Review · Reviewer_B5oV · 2024-10-18

**Soundness:** 3
**Presentation:** 1
**Contribution:** 1
**Rating:** 3
**Confidence:** 4

**Summary:**

This paper analyses the expressivity of GNNs with rational activation functions, building on previous work that characterises the expressivity of GNNs using a fragment of first-order logic (GC2). They provide two main results: 1) they show that there are GC2 expressions that cannot be uniformly (i.e. GNN size does not depend on input graph size) captured by GNNs with rational activations, and 2) they define a fragment of GC2 (RGC2) which GNNs with rational activations can capture. Finally, they run experiments on two artificial datasets (corresponding to two different queries) to test the abilities of Rational vs ReLU GNNs.

**Strengths:**

1. The problem statement and corresponding objectives are clear
2. There is good reference to existing work and how this paper fits into it
3. The paper builds upon an interesting body of work that is relevant to the ICLR community - the expressivity of GNNs

**Weaknesses:**

1. The significance of the contribution is small. It is not clear, either from references or experiments, why we should care about GNNs with rational activation functions - could such references or experiments perhaps be provided?. Thus, these (already minor) theoretical results about what fragments of logic they can and cannot express do not seem substantial.

2. The presentation is lacking.

2.1. The background takes up far too much space, with novel contributions of the paper first appearing on page 7. The background is also too verbose, since anyone who is interested in this work will either already be familiar with much of the background, or can go read some of the literature. I suggest rather summarising the background knowledge and referring the reader to the appendix / related work for a more in-depth explanation. For example, L185 - L205 uses a lot of space to define FOL on graphs. It can be condensed both by cutting whitespace, and by being less verbose.

2.2 The structure makes it difficult to follow: all theoretical results are presented in L335 - 352, with the proof sketches and explanations appearing far below. RGC2 is only defined in L416 but references in a theorem in L351. All technical terms should be defined before they are used and proof sketches / intuitions should appear near one another. Remark 5 is not useful, since it is referencing things that are only in the appendix, and cannot be made sense of without first reading the appendix. If it could be presented more intuitively, or in a way that makes it clearer why it is relevant to the proof, that would help.

2.3 In general, the space is used poorly, and the paper did not need the full 10 pages. Parts of the paper are lacking intuition and explanation (e.g. section 3) whilst others are far too verbose. For example, L250 - L269 ends up repeating much of what was said above.

3. The experiments are small in scope and the results that were obtained seem to undercut the significance of the theoretical contribution, since the theoretical advantage of CReLU GNNs does not transfer to a practical advantage. To better support the work, more substantial experiments should be performed across both artificial datasets (i.e. on more queries) as well as on some standard benchmarks.

**Questions:**

1. Why should we care about GNNs with rational activations?

2. What does L484 - L485 mean by "as the mean square error stabilizes around a value for large graphs"?

---

### Official Review · Reviewer_UZLz · 2024-11-01

**Soundness:** 2
**Presentation:** 3
**Contribution:** 2
**Rating:** 5
**Confidence:** 3

**Summary:**

This paper explores the impact of rational activation functions in specific classes of Graph Neural Networks (GNNs) on their capacity to uniformly approximate queries from the two-variable guarded fragment with counting (GC2). In this context, uniformly indicates that the number of GNN parameters is dependent solely on the query depth rather than the graph size. While prior studies show that GNNs with ReLU activations can uniformly express GC2, the authors demonstrate that certain GC2 queries cannot be captured by GNNs with rational activations. They further introduce RGC2, a fragment of GC2 expressible by GNNs employing rational activations. The authors conduct experiments demonstrating that approximations for specific queries remain stable as the number of nodes increases.

**Strengths:**

**Novelty and Idea:**

Given the demonstrated advantages of rational activation functions over ReLUs in traditional neural network architectures, the authors propose the compelling idea of extending this approach to Graph Neural Networks (GNNs).

**Theoretical Contribution:**

The authors demonstrate that, in the context of GC2, GNNs with rational activation functions exhibit lower expressivity compared to those using ReLU activations.

**Pedagogical Clarity:**

The authors have made effort to clarify the logical foundations required to understand GC2 and related fragments. Section 2, in particular, is highly accessible and does an excellent job of providing readers with the necessary background in logic. This careful exposition allows the paper to be more approachable to readers who may not have a strong background in formal logic.

**Weaknesses:**

**Lack of Literature Examples for GNN Types:**

 In Remark 1, the authors discuss three types of Graph Neural Networks (GNNs); It would be helpful if you could provide specific examples from the literature for each GNN type mentioned. This would give readers a better understanding of how these different GNN architectures have been applied in practice and studied theoretically.

**Clarity Issues in Example 2 of Section 3:**

 Example 2 in Section 3 presents some ambiguities, potentially due to complex logical reasoning. For readers who may not be experts in logic, further clarification or simplification would enhance comprehension. Presenting the example in more accessible terms could improve the paper’s accessibility and impact.

**Figure 1 Caption:**

To improve Figure 1, consider expanding the caption to provide more context about what the figure represents and how it relates to the main concepts discussed in the paper. This would allow readers to understand the figure's significance without needing to refer back to the main text.

**Limited Scope of Experiments:**

 The experimental evaluation in this paper is restricted to two queries, and the results do not fully corroborate the theoretical assumptions made by the authors. Additionally, the authors claim that a ReLU GNN can approximate CReLU GNN, but no direct comparison is provided in the experiments. Including a comparison with a ReLU GNN would strengthen the experimental results and help substantiate the authors’ claims.

**Questions:**

**Universe $A$ Notation:**

 In graph theory, the symbol $A $ typically denotes the adjacency matrix representing the structure of a graph. However, throughout the paper, $ A $ appears to represent the universe and is set to $ V $ in each definition. Could you clarify whether the choice of $ A $ here is tied to a general logic definition? Additionally, in Example 2 of Section 3, $ A $ is introduced as a matrix. To avoid confusion, would it be preferable to use a different letter for this matrix?

**Clarification in Example 2 of Section 3:**

 I would appreciate additional clarification on Example 2 in Section 3. Specifically:
   - **Usage of Vector $ c $**: The vector $ c $ is defined, but it appears to go unused. Could you elaborate on its intended purpose?
   - **Indices in $ \xi^4(G, v)_i $**: The notation $ \xi^4(G, v)_i $ introduces an index $ i $ whose meaning is unclear. As this may pertain to logical constructs that not all readers are familiar with (like me), additional explanation here would improve clarity.

**Typographical Error in Proof Overview (Line 444):**

In the proof overview, line 444, there appears to be a typographical error. The text suggests that rational GNNs can uniformly express GC2, which is contradictory to Theorem 3. Could you confirm if this is an error or provide clarification?

**Loss Curve Analysis:**

Have you analyzed the training loss curves for both CReLU and rational GNNs on the training set? Examining these curves could provide insight into your theoretical claim. For instance, if rational GNNs indeed struggle to express certain GC2 queries, we might observe a higher or stagnant training loss. Such empirical observations could help substantiate your theoretical findings.

**General typographical error:**

 - **line 181**: for **a** some elements of comparison of expressivity of targeted vs standard ones. **a** should be removed.
 - **line 309** :  $(Q_1,Q_2,Q_3,Q_4)$ instead of $(Q1,Q2,Q3,Q4)$.
 - **line 360** : Our approach **and** can easily extend to a large family of GC2 queries. **and** should be removed.
 - **line 450** : **Then**, we **then**. One **then** should be removed.

---

### Official Review · Reviewer_ftCq · 2024-11-02

**Soundness:** 3
**Presentation:** 3
**Contribution:** 2
**Rating:** 5
**Confidence:** 3

**Summary:**

This paper studies the expressivity of graph neural networks from the perspective of logic. Concretely, the paper focus on investigating the uniform expressivity of the specific variant of GNNs namely Message Passing Neural Networks. Existing results in [Barcelo &. Al., 2020] have shown that GC2 can be expressed uniformly by MPNNs equipped with ReLU activations, however, uniform expressivity of MPNNs with some other activations, such as rational activations, still remains uninvestigated. This paper investigates the uniform expressivity of MPNNs equipped with rational activations, and proposes the fragment of GC2 uniformly captured by them. The results are justifies by both theoretical analysis and numerical experiments.

**Strengths:**

- This paper extends the results of [Barcelo &. Al., 2020] to MPNNs equipped with rational activations. Many previous works on GNN expressivity do not consider the specific design of aggregation / combination functions and assume they can be arbitrary function. It is therefore necessary and practical to study what GNNs, equipped with different activation functions, can uniformly express.
- For MPNNs with rational activations, this paper adequately shows:
  - Rational MPNNs cannot uniformly express GC2 formulas (which are uniformly expressed by MPNNs if we don't restrict the activation functions)
  - There's a subset of GC2 which is uniformly expressed by rational MPNNs.

**Weaknesses:**

- The main contribution of this work is to point out RGC2 as a subset of GC2 which discribes the uniform logical expressivity of rational MPNNs. However, there is only one theorem (Theorem 4) that states all RGC2 formulas are expressed by rational MPNNs, and for a more complete study of GNN expressivity, it is still necessary to:
  - Show RGC2 completely captures the logical expressivity of rational MPNNs (if this was the case). This can be done by either:
    - Proving that RGC2 is equivalent to rational MPNNs in terms of distinguishability, i.e. for arbitrary graphs $G,H$, rational MPNNs distinguish $G$ and $H\iff$ there is a RGC2 formula that distinguish $G,H$ (which I believe is not the case)
    - Proving that RGC2 is the largest logic set captured by rational MPNNs, i.e. for arbitrary logic formula $Q\notin$RGC2, $Q$ is not uniformly captured by rational MPNNs.
  - If the above statements do not hold for RGC2, discuss why it is not feasible to find the subset of logic formulas that satisfies the above statements, and explain why RGC2 is large enough to cover up important properties of rational MPNNs.

## Suggestions:
- This paper focus on a very specific setting of GNNs: MPNNs with rational activations. This paper can be further strengthened by:
  - Discussing why it is important to study rational activations.
  - Extending to more complex GNN variants (i.e., consider higher-order MPNNs, which are direct extensions of MPNNs and are expressed by $\mathrm{FOC}_k$).
  - Extending to different activation functions.

- Other suggestions:
  - The discussion of RGC2 as the main contribution of the paper is short. For example, the authors can discuss the general difference between GC2 and RGC2 (i.e. what kinds of logic formulas are discarded, what patterns GC2 can capture but RGC2 cannot).
  - Theorem 4 is the main theorem of this paper and Section 5 introduces the main constribution. Maybe consider putting Theorem 4 in Section 5?

**Questions:**

Please refer to Weakness. Among them, the first one is the most important.

---

### Official Review · Reviewer_LSpf · 2024-11-12

**Soundness:** 4
**Presentation:** 4
**Contribution:** 2
**Rating:** 5
**Confidence:** 3

**Summary:**

The authors investigate logical expressivity of GNNs, i.e., the first order logic classifiers that can be captured by a GNN, in the uniform setting, i.e., the graph sizes can vary but the parameter size of the GNN remains fixed. This has been completely answered for GNNs without a global readout by Barcelo et al. (2020) --- showing that all such classifiers are exactly the once expressible in the graded fragment of first order logic with two variables, and counting quantifiers (GC2).
In this paper, the authors look at the expressivity of a restricted setting, i.e., the activations functions are represented by a rational function.
- The authors show that not all classifiers  in GC2 can be captured by a rational GNN.
- They introduce a restricted fragment of GC2, called RGC2, that can be captured by the GNNs.

**Strengths:**

- Clarity. The paper is clearly written. I have not checked the proofs provided in the appendix. But the paper draws very clear and well-motivated connections to Barcelo et al. and does an excellent job at presenting the results.

**Weaknesses:**

- Novelty and relevance. My main concern is the novelty and the relevance of the results. NNs with rational activations seems really a niche, and I am unable to understand the motivation to investigate this fragment.
- I maybe mistaken, but it also seems that RGC2 is an arbitrary restriction picked up by authors to show some positive results. I think a better attempt towards providing positive results for a larger fragment, or even completely capturing Rational GNNs could have been attempted.
- I would have been more inclined towards acceptance, if a clear picture of why RGC2 is the best fragment they can capture would have been presented.

**Questions:**

- Could you please briefly explain, and maybe point to the parts of proof, that make capturing anything larger than RGC2 hard?
- Or otherwise, what motivates you to conjecture " We conjecture it is not the case and that RGC2 is close to the logic expressed by rational GNNs"?

---

### Note · Authors · 2024-11-18

**Comment:**

We sincerely thank the reviewers for their insightful feedback, which will enable us to make substantial improvements to the content of our article. Given its current state, we believe it is not ready for publication yet, and have decided to retract the article in order to implement these changes.

**Withdrawal Confirmation:**

I have read and agree with the venue's withdrawal policy on behalf of myself and my co-authors.